# Community surveys of the prevalence, distribution, and coinfection of helminth and protozoan infections in semiurban and rural areas of Gabon, Central Africa

Jean Claude Dejon-Agobé[1,2]*, Christian Chassem-Lapue[1],
Paul Alvyn Nguema-Moure[1], Moustapha Nzamba Maloum[1], Roméo-Aimé Laclong Lontchi[1],
Mahmoudou Saidou[1], Ynous Djida[1], Jean Ronald Edoa[1,3], Yabo Josiane Honkpéhèdji[1,4],
Jeannot Fréjus Zinsou[1,2], Bayodé Roméo Adégbitè[1,2,3], Martin Peter Grobusch[1,2,3,5],
Peter Gottfried Kremsner[1,2,5], Ayôla Akim Adegnika[1,2,4,5]

1 Centre de Recherches Médicales de Lambaréné, Lambaréné, Gabon, 2 Institut für Tropenmedizin, Universitätsklinikum Tübingen, Tübingen, Germany, 3 Center of Tropical Medicine and Travel Medicine, Department of Infectious Diseases, Division of Internal Medicine, Amsterdam University Medical Centers, Location AMC, University of Amsterdam, Amsterdam, The Netherlands, 4 Leiden University Medical Center for Infectious Diseases, Leiden, Netherlands, 5 German Center for Infection Research (DZIF), Partner Site Tübingen, Tübingen, Germany

* jcagobe@gmail.com, jcagobe@cermel.org

## Abstract

### Background

Protozoa and helminths cause significant morbidity and mortality, particularly in tropical and subtropical regions where an accurate understanding of their epidemiological profile is needed to improve their control. In Gabon, a country endemic for a diverse range of both helminths and protozoa, epidemiological data for protozoa are lacking, whereas updated data for helminths are needed. This study aimed to describe the distribution of helminth and protozoan infections in the Moyen-Ogooué province of Gabon.

### Methods

This cross-sectional study included individuals aged one year and older living in the study areas for at least one year. The participants were selected via a stratified sampling procedure. Blood, urine, and stool samples, along with sociodemographic data, were collected. Soil-transmitted helminths (STHs) were diagnosed using the Kato-Katz, coproculture and Harada-Mori techniques. Urogenital schistosomiasis was diagnosed using the urine filtration technique. Intestinal protozoa were diagnosed using the mercurothiolate-iodine-formol technique. *Plasmodium* spp. and filarial infections were diagnosed by thick blood smear microscopy, and, in addition for filaria, by leucoconcentration technique.

**Data availability statement:** All data are in the manuscript and/or Supporting Information files.

**Funding:** This work was supported by the European and Developing Countries Clinical Trials Partnership (CSA2020NoE 3100 to AAA) and the German Center for Infection Research (project TTU 03.707-03 to AAA). The funders had no role in study design, data collection and analysis, decision to publish, or preparation of the manuscript.

**Competing interests:** The authors have declared that no competing interests exist.

## Results

A total of 1,084 participants were included, with a mean age of 31.6 years (SD: 23.6) and a female-to-male sex ratio of 1.15. The overall prevalence of helminth infections was 36% (95%IC: 33–39), with STHs being most common (21%; 95%CI: 18–23), followed by schistosomiasis (11%; 95%CI: 8 – 13) and filariasis (9%; 95%CI: 7–10). The most prevalent STH species were *Trichuris trichiura* (11%; 95%CI: 10–14), followed by hookworm (9%; 95%CI: 8–11). The prevalence of *Plasmodium* spp. was 13% (95%CI: 11–15), and the overall prevalence of intestinal protozoa was 28% (95%CI: 25–31), with *Blastocystis hominis* (11%; 95%CI: 9–13) and *Entamoeba coli* (8%; 95%CI: 7–10) being the most common intestinal protozoan species. Coinfections with multiple parasite species were observed in 42% of the infected participants, predominantly involving *T. trichiura*, *Schistosoma haematobium*, and *Plasmodium* spp. infection prevalence varied with age, gender, location, and occupation.

## Conclusion

This study reveals a moderate prevalence of helminths and protozoa in our community, with age, gender, and location playing a significant role in their distribution, as do common coinfections between helminths and protozoa. These findings call for further research to provide valuable insights for controlling helminth transmission in the region.

## Author summary

Helminths and protozoan infections pose major public health problems worldwide, particularly in tropical and subtropical areas where the climate and living conditions are favourable for their development. For a better estimate of their morbidity and better control of helminths in particular, epidemiological data are necessary for each endemic area. This information should help identify the impact of these infections in communities and establish more appropriate health policies. Gabon, a country in Central Africa, is endemic to both helminths and protozoa. While some epidemiological data on helminths exist, although they are sometimes old, epidemiological data on protozoa are rare. To fill this gap, we conducted a cross-sectional survey in Moyen-Ogooué, one of Gabon's nine provinces, during which we were able to randomly include 1,084 participants aged one year and over. Blood, stool and urine samples were obtained for the diagnosis of endemic helminths and protozoa via reference methods. Our results revealed that the prevalence of helminths and protozoa is moderate in Moyen-Ogooué Province and that their morbidity is linked to factors such as locality, occupation, age and gender, depending on the species of the parasite. Overall, these data highlight the need to implement control strategies adapted to the realities of our study area, which could, as we strongly believe, improve the health of these populations.

## Background

Human parasites are microorganisms that can be categorized into two groups: endoparasites, which include protozoa and helminths, and ectoparasites, such as scabies, lice and ticks. Parasitic infections caused by protozoa and helminths can affect either the intestines, other organs and tissues, and/or the bloodstream. Because they are responsible for high morbidity and mortality worldwide [1], endoparasitic infections represent a significant global public health concern. It is estimated that more than three billion people are infected with intestinal parasites globally [2], with soil-transmitted helminths (STHs) being the most prevalent, affecting roughly 25% of the world's population [3]. In terms of protozoa, the 2023 World Malaria Report indicates that, in 2022, there were up to 249 million cases of malaria [4]. Tropical and subtropical regions, including the Asia-Pacific region, South America, and Africa, are disproportionately affected by these infections [1,5]. Schistosomiasis, which is endemic in many of these regions, is considered the second most important parasitic disease after malaria [6]. In Central and Western sub-Saharan Africa, at least 10 million people are infected with the filarial worm *Loa loa* [7], and when Latin America is included in the latter region, an estimated 100 million people are affected by mansonellosis [8].

Helminths are invertebrates with elongated, flat, or round bodies. They are divided into three main groups on the basis of the morphology of the adult worm: flatworms or platyhelminths, which may be segmented (cestodes) or not (trematodes or flukes); and roundworms (nematodes). Adult helminths may reside in the gastrointestinal tract (nematodes such as STH), blood vessels (trematodes such as schistosomes), and lymphatic system (nematodes such as *Wuchereria bancrofti*) or subcutaneous tissues (nematodes such as *Loa-loa*) [9]. In contrast, protozoa are single-celled microscopic organisms capable of replicating in the host. Intestinal protozoa are typically transmitted through faecal contamination, whereas protozoa living in blood or tissues are transmitted via arthropod vectors, such as mosquitoes or sandflies. Pathogenic protozoa are classified into four groups on the basis of their mode of movement: Sarcodina or amoebae (*Entamoeba* spp. *Endolimax nana, Iodamoeba butschlii, Acanthamoeba* spp.), Mastigophora or flagellates (*Giardia duodenalis, Leishmania* spp.*, Trypanosoma* spp.*, Chilomastix mesnili, Dientamoeba fragilis, Trichomonas vaginalis*), Ciliophora or ciliates (*Balantidium coli*) and Apicomplexa also known as sporozoa (*Plasmodium* spp., *Cryptosporidium hominis, Toxoplasma gondii, Cystoisospora belli, Cyclospora cayetanensis, Sarcocystis hominis*) [9].

Severe infections caused by helminths and protozoa can lead to significant morbidity, including anaemia, malnutrition, stunted growth, and neurocognitive impairments, which affect children's cognitive and adult productivity [5,10]. Schistosomiasis can present in intestinal or urogenital forms, with the former potentially causing severe liver damage and the latter linked to urogenital damage [11]. Hookworms, which infect between 500 and 800 million people annually, cause significant blood loss, which is particularly harmful to women of reproductive age (WRA) [12]. In terms of mortality, *Entamoeba histolytica* is responsible for 40,000–100,000 deaths each year, making it one of the deadliest parasitic infections globally [5]; second to malaria, which claimed 608,000 lives in 2022, [4]. To reduce the morbidity and mortality associated with these infections, the World Health Organization (WHO) has developed an integrated approach for helminth control involving chemoprevention; improved water, sanitation and hygiene (WASH) services; behavioural changes; vector control; and the development of new drugs and vaccines [13]. Despite these efforts, parasitic infections remain highly prevalent in tropical and subtropical regions.

Gabon, a Central African country, is endemic for helminths such as filariasis caused by *Onchocerca volvulus* [14], *Loa loa* or *Mansonella perstans* [7,15], schistosomiasis [16], STHs [17], as well as intestinal or blood protozoa, notably *Plasmodium* spp. [17,18]. However, the available data on the prevalence of these parasitic infections are either scarce or outdated. This study aims to provide actual epidemiological data on the distribution of helminth and protozoan infections in Gabon.

## Methods

### Ethics statement

The study protocol was approved by the Centre de Recherches Médicales de Lambaréné (CERMEL) scientific committee (SRC2021–24), the CERMEL institutional (CEI-003/2022) and the national (026/2022/CNE/SG/P) ethics committees. Before data or sample collection, the study was explained to participants, including the purpose, potential benefits, risks,

and procedures to obtain their consent to participate. For participants under 18 years of age, written informed consent was granted by their parents or legal representatives. Those aged 12–17 years also gave their assent to participate. The study adhered to the principles of Good Clinical Practice from the International Conference on Harmonization [29] and the Declaration of Helsinki [30].

## Study site and study areas

The study was conducted at CERMEL [19] and took place in the Ogooué et des Lacs department, one of the two departments of the Moyen-Ogooué province in Gabon. As shown in Fig 1, Moyen-Ogooué Province is the smallest of Gabon's nine provinces in terms of area, yet the sixth most populous, with 69,287 inhabitants [20]. The province is traversed by the Ogooué River, the country's primary river, which is 1,200 km long and creates a landscape of numerous lakes, ponds, rivers, swamps, and marshes. Moyen-Ogooué province is bisected by the equator and experiences a hot and humid equatorial climate, with two rainy seasons (November to mid-December and February to June) and two dry seasons (July to October and mid-December to mid-January). The rainforest covers 84% (15,500 km²) of the province. Lambaréné, the provincial capital, is a semiurban area with a population of 44,000. The N1 national road, which crosses Lambaréné, connects the city to Libreville (the capital of the country) in the northwest (240 km) and to towns in the southern part of the country [21]. For this study, the area along the N1 road between Lambaréné and the province's northwestern border was designated the northern rural area, and the area between Lambaréné and the southern border was designated the southern rural area.

## Study design and study population

This was a cross-sectional study conducted between September 2022 and September 2023. All individuals aged one year and older living in the study areas for at least one year were invited to participate.

## Sample size calculation

This study aimed to determine the prevalence of helminth and protozoan infections in the study area. To calculate the required sample size, we performed a calculation to obtain the maximum possible sample size that would be valid for all the species investigated. Therefore, using Daniel's sample size calculation formula in cross-sectional studies ($n = Z^2 \times P \times (1 - P)/d^2$) and considering a prevalence "P" of 50%, with a desired precision "d" of 0.05 and a 95% confidence interval (given a z-score of 1.96), the minimum sample size "n" to be included was 385 participants, and was considered for Lambaréné and both rural areas, leading to a minimum of 770 participants being required for the study.

## Sampling procedure

The participants were selected via a three-stage stratified sampling method. First, 24 neighbourhoods in Lambaréné (semi-urban area) and 24 surrounding villages (rural areas) were randomly chosen from a preestablished list. In each selected village or neighbourhood, one out of every three houses was chosen, starting from a predetermined point (e.g., the village or neighbourhood entrance, the main crossroads in the neighbourhood). If a selected house was uninhabited or closed, the next house was selected. Finally, in each household, a number was assigned to each family member aged one year and older with the agreement of the head of the family, and four individuals were chosen by drawing lots. In households with four or fewer residents, all individuals were invited to participate. Only volunteers who were randomly selected and who provided written informed consent were included in the study.

## Study procedure

Once signed informed consent was obtained, study nurses recorded the volunteers' parameters (temperature, weight, height), conducted interviews on sociodemographic information, and collected blood samples. Global positioning system

PLOS Neglected Tropical Diseases

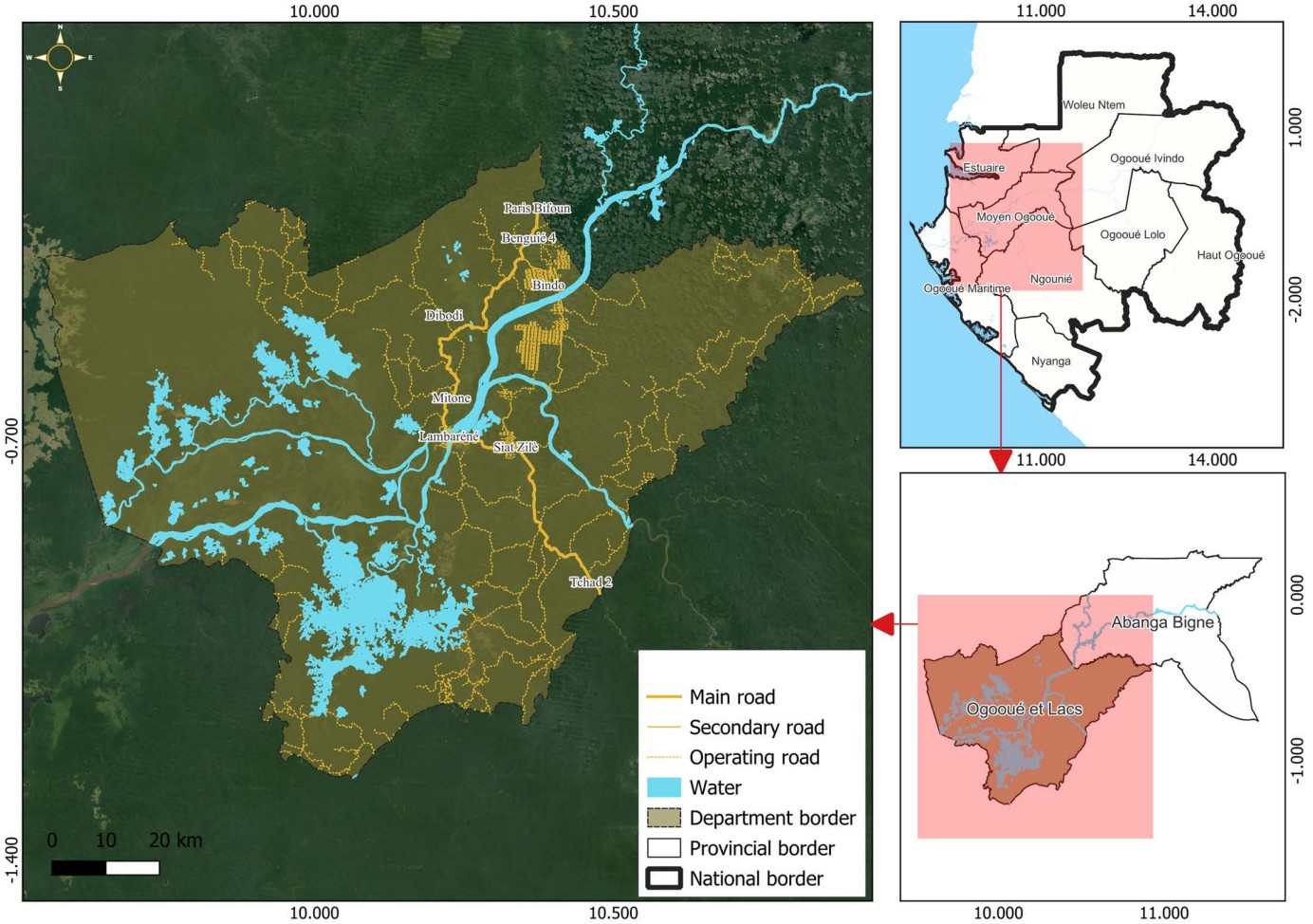

**Fig 1. Map of Gabon highlighting the study area.** The base layer for the country and the shape of the national borders were obtained from the 2018-2022 GADM database, an open licence (https://gadm.org/download_country.html). QGIS version 3.34.11 was used to create the map.

(GPS) coordinates were collected for each participant, and plastic containers were provided for participants to collect urine samples over three consecutive days unless the first or second sample was positive for the presence of *S. haematobium* eggs. In that case, the remaining sample(s) was (or were) not required anymore, and a stool sample was collected the day after the visit or as soon as possible. Field teams collected these samples every morning between 8:00 am and 11:00 am and transported them to the laboratory, where they were immediately stained for approximately ten minutes and then read or cultured. The results of the parasitological and biological tests were recorded in the participants' case report forms (CRFs) before digitization. Thick blood smear (TBS), haematology and stool results were communicated to the participants or their legal representatives. Positive cases were treated by the study physician, with praziquantel 40 mg/kg of body weight once for schistosomiasis, albendazole (ABZ) 400 mg once daily for three consecutive days for STH infections, and artemether-lumefantrine dosed according to the participant's age for malaria. For other diagnosed infections, a prescription was provided by the study physician, or participants were referred to an appropriate health centre if needed.

## Laboratory analysis

The detection of *Plasmodium spp.* in blood was performed via microscopy using TBS prepared according to the Lambaréné method described elsewhere [22]. Briefly, 10µL of blood was evenly distributed over a rectangular area of 10x18 mm, dried, and stained. The number of parasites was counted for at least 100 high-power fields and parasitaemia per microlitre was calculated [22]. Similarly, filarial infections were detected via TBS using the Lambaréné method and, in addition, via the leucoconcentration technique, which consists of mixing one ml of venous blood in a tube with one ml of 2% saponin and one ml of physiological liquid. The mixture was left to stand for five min at room temperature and then centrifuged at 2000 rpm for 10 min. The sediments were then transferred between the slide and coverslip and read under a microscope at Gx10 [23]. The Kato–Katz method, which consists of examination via light microscopy 10 min after the preparation of a smear performed with a stool sample as described elsewhere [23], was used to detect *A. lumbricoides*, *T. trichiura*, and hookworm eggs in the stool samples, whereas the coproculture and Harada–Mori techniques [23] were used in parallel for the detection of hookworm and *Strongyloides stercoralis* larvae in the stool samples. Briefly, approximately one gram of stool sample was incubated at 22–28°C for seven or ten days, respectively, to allow larvae to grow which can then be observed under a microscope as described elsewhere [23]. Mercurothiolate-iodine-chloroform staining was used to detect of intestinal protozoa cysts in fecal samples via direct microscopic examination as described elsewhere [23]. The urine samples were analysed for the presence of *S. haematobium* eggs using the urine filtration technique. which consists, as described elsewhere, of passing 10 ml of fresh urine through a 12 µm microfilter membrane (Whatman type) via a syringe. The membrane was then transferred onto a glass slide, mounted on a microscope and read using a low-power objective (10×) of a light microscope [24].

## Statistical consideration

Data were collected via a CRF and digitalized using the electronic data capture tool REDCap [25] hosted at CERMEL. The final database (see S1 Dataset) was exported to R software version 3.2.4 for statistical analysis. Age was analysed as both a continuous variable and a categorical variable. For the categorization of age, we used the following definitions for children: toddlers (ages 1–2 years) and preschoolers (ages 3–4 years) who were combined into one age group; school-aged children (ages 5–12 years) and teenagers (ages 13–19 years) [26]. For statistical reasons, participants aged 20 and over were grouped into groups of 20–49 years and 50 years and over, respectively. Women of reproductive age (WRA) was defined as any woman between the ages of 15 and 49 [27]. Participants were considered positive and classified as infected if at least one egg or larva was detected in a stool sample or if parasites were identified in a blood smear. The intensity of STH infections was categorized as light, moderate or heavy, whereas the intensity of schistosomiasis was categorized as light or heavy using a threshold indicated elsewhere [28]. Quantitative variables were summarized using means and standard deviations (SDs) for normally distributed data or medians and interquartile ranges for nonnormally distributed data. Proportions and 95% confidence intervals (CIs) were used to summarize the qualitative data. The binom exact function of the Binom R package was used to calculate the 95% CI. The difference in proportions was determined under the assumption that no overlapping confidence intervals indicate statistical significance.

## Results

### Characteristics of the study population

Of the 1092 volunteers who were randomly selected and invited to participate in the study, six refused to participate, whereas two were not able to provide any samples. Therefore, a total of 1084 participants were included in the study, with a mean (±SD) age of 31.6 (± 23.6) years and a female-to-male ratio of 1.15. Among the 579 females included in the study, 36% were WRA. Our study population was mainly students, especially those from primary and secondary schools (35%), people with no income-generating occupation (28%) or farmers (24%), with 44% of them living in Lambaréné. The demographic characteristics of the study population are summarized in Table 1. A total of 115 participants presented with an ancillary temperature higher than 37.5°C during the visit, while 342 complained of health problems, which mainly included

fever or a history of fever over the past three days and pain (headache, chest pain, arthralgia, lumbalgia, etc.). Seventeen participants reported being on ongoing antimalarial treatment on the day of their inclusion in the study.

## Prevalence of helminth infections

As depicted in Fig 2, a total of 357 participants were infected with helminths, resulting in a prevalence of 36% (357/980, 95%CI: 33–39). As presented in Tables 2 and 3, STH infections were mostly prevalent at 21% (202/980, 95%CI: 18–23), followed by schistosomiasis at 11% (115/1084, 95%CI: 8 – 13) and filariasis at 9% (94/1084, 95%CI: 7–10). Among the STH species, *T. trichiura* was the most prevalent species, affecting 11% (113/980, 95% CI: 10–14) of the participants, followed by hookworm at 9% (92/980, 95%CI: 8–11), *A. lumbricoides* at 5% (45/980, 95%CI: 3–6), and finally *S. stercoralis* at 3% (26/980, 95%CI: 2–4). Of the 91 participants positive for hookworm, 10 were only positive by the presence of larvae. In terms of filariasis, *L. loa,* with 7% prevalence (76/1076, 95%CI: 6–9), and *M. perstans,* with 3% prevalence (30/1076, 95%CI: 2–4), were detected (Table 3). Only two cases of *Enterobius vermicularis* infection from both rural areas, both male of 55-year-old farmer from the northern rural area and 9-year-old student from the southern rural area; and one case of tapeworm infection (*Taenia solium*) was detected in the northern rural area (Fig 2).

## Distribution of STH infections

Table 2 presents the distribution of STH infections by age, gender, location, and occupation. The prevalence of *A. lumbricoides* was low (≤ 4%) across all age groups, except for a slight peak of 8% (95% CI: 6–12) in the 5–19 age group. For *T. trichiura*, the prevalence was relatively consistent across age groups, ranging from 7% to 13%. The number of

**Table 1. Study participant characteristics.**

|  | Study population | |
|---|---|---|
|  | n | % |
| **N (overall population)** | 1084 | – |
| **Age (mean, ± SD)** | (31.6, ± 23.6) | – |
| **Age group** | | |
| 1 – 4 | 109 | 10.0 |
| 5 – 19 | 345 | 31.8 |
| 20 –49 | 315 | 29.1 |
| ≥ 50 | 315 | 29.1 |
| **Gender** | | |
| Female | 579 | 53.4 |
| Women of Reproductive Age | 208 | 35.9 |
| Male | 505 | 46.6 |
| Female-to-male sex ratio | (1.15) | |
| **Location** | | |
| Lambaréné | 483 | 44.6 |
| Southern rural area | 223 | 20.6 |
| Northern rural area | 378 | 34.9 |
| **Occupation** | | |
| No occupation | 298 | 27.5 |
| Farmers | 259 | 23.9 |
| Students | 384 | 35.4 |
| Others | 143 | 13.2 |

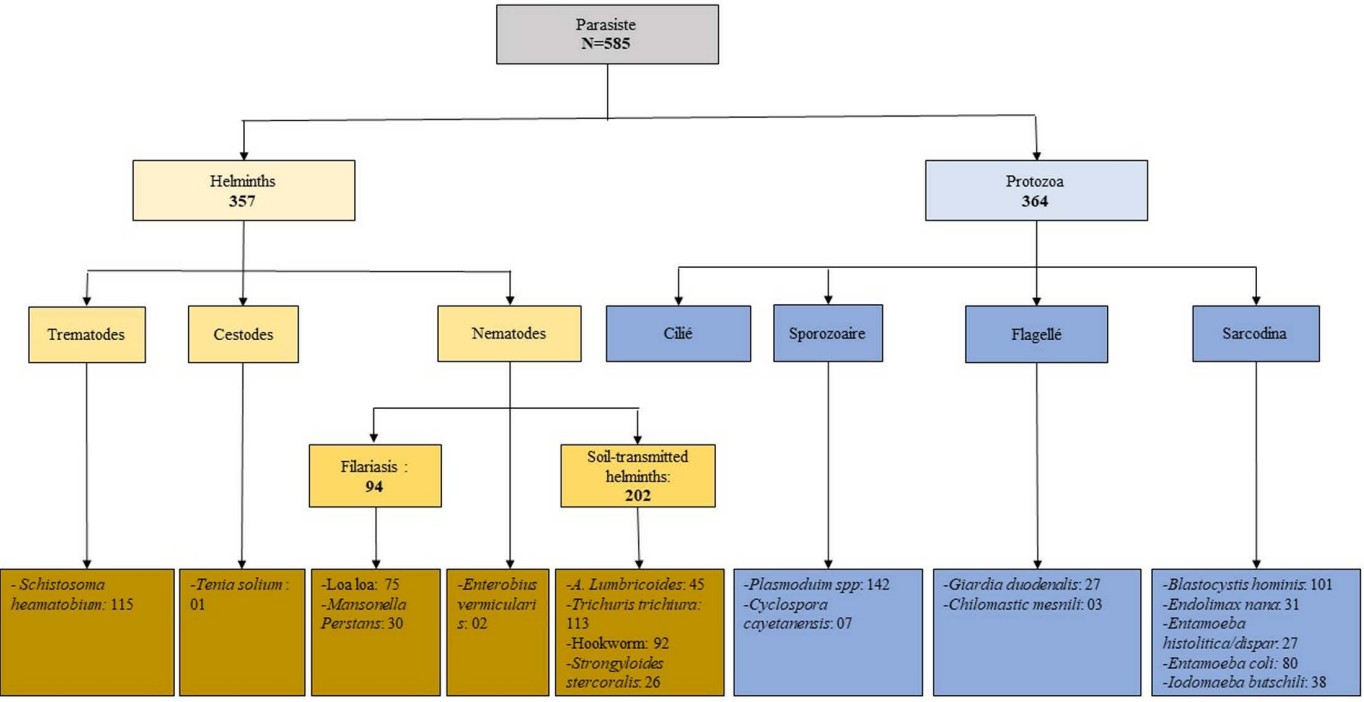

**Fig 2. Flow of parasite species found during the survey.**

hookworm infections increased with age from only one case observed in the 1–4-year-old age group to 16% (95% CI: 12–21) among participants aged 50 years and over, whereas no cases of infection were found in the 1–4-year-old age group for *S. stercoralis,* with no clear pattern observed across the other age groups. Males had higher infection rates than females did for *T. trichiura* (14%, 95% CI: 11–18 *vs* 9%, 95% CI: 7–12), hookworm (13%, 95% CI: 10–17 *vs* 6%, 95% CI: 4–8), and *S. stercoralis* (4%, 95% CI: 2–6 *vs* 1%, 95% CI: 1–3). The prevalence of helminth infection was lower in Lambaréné than in both rural areas. A significantly greater prevalence of *T. trichiura* was observed in northern rural areas than in southern rural areas (24%, 95 CI: 19–29 *vs* 11%, 95% CI: 8–16). Farmers had a significantly greater prevalence of STH infections (35%, 95% CI: 29–41) than the other occupational groups did, and this pattern was true for *T. trichiura* (16%, 95% CI: 12–21) and hookworm (24%, 95%CI: 19–30) when considering STH species.

## Distribution of filariasis

Table 3 presents the prevalence of filariasis by age, gender, location, and occupation. Cases of *Loa loa* infection were observed first at the age of 4, and the number of cases increased with age; whereas *M. perstans* infection was observed first from participant aged 19-year-old, and the number of cases was similar across the two other age groups of 20–49 years and ≥ 50 years. Compared with females, males had slightly higher prevalence rates for *Loa-loa* infection (5%, 95%CI: 3–7 *vs* 9.4%, 95%CI: 6–11) and *M. perstans* (2%, 95%CI: 1–3 *vs* 4%, 95%CI: 3–6). *Loa loa* and *M. perstans* infections were significantly more prevalent in southern rural areas than in northern rural areas (14%, 95% CI: 10–19 *vs.* 7%, 95% CI: 4–10 and 7%, 95% CI: 4–11 *vs* 3%, 95% CI: 1–5) and Lambaréné (14%, 95% CI: 10–19 *vs.* 4%, 95% CI: 2–6, and 7%, 95% CI: 4–11 *vs.* 1%, 95% CI: 0–6). Farmers (14%, 95% CI: 10–19) had statistically significant higher *L. loa* prevalence compared to participant with no income-generating occupation (*vs.* 6%, 95% CI: 4–9) and students (*vs.* 2%, 95% CI: 1–4).

**Table 2. Prevalence and distribution of soil-transmitted helminths (STHs) among the 980 study participants who provided stool samples.**

| | N | Any STH | | Ascaris lumbricoides | | Trichuris trichiura | | Hookworm | | Strongyloides stercoralis | |
|---|---|---|---|---|---|---|---|---|---|---|---|
| | | n, (%) | 95%CI | n, (%) | 95%CI | n, (%) | 95%CI | n, (%) | 95%CI | n, (%) | 95%CI |
| **Overall** | 980 | 202, (20.6) | 18.1–23.3 | 45, (4.6) | 3.4–6.1 | 113, (11.5) | 9.6 – 13.7 | 92, (10.4) | 8.6 – 12.5 | 26, (2.6) | 1.7 – 3.9 |
| **Age range** | | | | | | | | | | | |
| 1–4 | 89 | 9, (10.1) | 4.7 – 18.3 | 3, (3.4) | 0.0 – 19.0 | 6, (6.7) | 2.5 – 14.1 | 1, (1.1) | 0.0 – 6.1 | 0, - | – |
| 5–19 | 312 | 63, (20.2) | 15.9 – 25.1 | 26, (8.3) | 5.5 – 12.0 | 40, (12.8) | 9.3 – 17.0 | 17, (5.4) | 3.2 – 8.6 | 7, (2.2) | 0.9 – 4.6 |
| 20–49 | 280 | 55, (19.6) | 15.1 – 24.8 | 5, (1.8) | 0.6 – 4.1 | 34, (12.1) | 8.6 – 16.5 | 25, (8.9) | 5.9 – 12.9 | 10, (3.6) | 1.7 – 6.5 |
| ≥ 50 | 299 | 75, (25.1) | 20.3 – 30.4 | 11, (3.7) | 1.8 – 6.5 | 33, (11.0) | 7.7 – 15.1 | 49, (16.4) | 12.4 – 21.1 | 9, (3.0) | 1.4 – 5.6 |
| **Gender** | | | | | | | | | | | |
| Female | 536 | 91, (17.0) | 13.6 – 20.0 | 25, (4.7) | 3.0 – 6.8 | 50, (9.3) | 7.0 – 12.1 | 33, (6.2) | 4.3 – 8.5 | 10, (1.9) | 0.9 – 5.3 |
| WRA* | 190 | 22, (11.6) | 7.4 – 17.0 | 5, (2.6) | 0.8 – 6.0 | 12, (6.3) | 3.3 – 10.8 | 4, (2.1) | 0.6 – 5.3 | 4, (2.1) | 0.6 – 5.3 |
| Male | 444 | 111, (25.0) | 21.0 – 29.3 | 20, (4.5) | 2.8 – 6.9 | 63, (14.2) | 11.1 – 17.8 | 59, (13.3) | 10.3 – 16.8 | 16, (3.6) | 2.1 – 5.8 |
| **Location** | | | | | | | | | | | |
| Lambaréné | 427 | 29, (6.8) | 4.6 – 9.6 | 5, (1.2) | 0.4 – 2.7 | 17, (4.0) | 2.3-6.3 | 12, (2.8) | 1.5-4.9 | 3, (0.7) | 0.1-2.0 |
| Southern rural area | 197 | 66, (33.5) | 27.0-40.6 | 20, (10.1) | 6.0—15.2 | 26, (13.2) | 8.8-18.7 | 31, (15.7) | 10.9-21.6 | 8, (4.1) | 1.8-7.8 |
| Northern rural area | 356 | 107, (30.1) | 25.3-35.1 | 20, (5.6) | 3.5 – 8.5 | 70, (19.7) | 15.7-24.2 | 49, (13.8) | 10.4-17.8 | 15, (4.2) | 2.4-6.9 |
| **Occupation** | | | | | | | | | | | |
| No occupation | 310 | 45, (14.5) | 10.8-19.9 | 11, (3.5) | 1.8-6.3 | 21, (6.8) | 4.2-10.2 | 16, (5.2) | 3.0-8.2 | 9, (2.9) | 1.3-5.4 |
| Farmers | 238 | 82, (34.5) | 28.4 – 40.9 | 10, (4.2) | 2.0 – 7.6 | 38, (16.0) | 11.6 – 21.3 | 57, (23.9) | 18.7 – 29.9 | 9, (3.8) | 1.7 – 7.1 |
| Students | 351 | 63, (17.9) | 14.1 – 22.4 | 23, (6.6) | 4.2 – 9.7 | 46, (13.1) | 9.8 – 17.1 | 13, (3.7) | 2.0 – 6.3 | 8, (2.3) | 1.0 – 4.4 |
| Others | 81 | 12, (14.8) | 7.9 – 24.4 | 1, (1.2) | 0,0 – 6.7 | 8, (9.9) | 4.4-18.5 | 6, (7.4) | 2.8-15.4 | 3, (3.7) | 0.8 – 10.4 |

*Women of reproductive age

## Distribution of schistosomiasis

Table 3 presents the prevalence of schistosomiasis by age, gender, location, and occupation. The infection cases started at two years of age, with the highest prevalence observed among those aged 5–19 years: 20% (95% CI: 16–25). Although individuals of all age groups were infected, the ≥ 50 years age group exhibited the lowest prevalence, 3% (95% CI: 2–6). No statistically significant gender difference was observed (13%, 95% CI: 10–16 *vs* 10%, 95% CI: 8–13). The prevalence of the infection was significantly higher in Lambaréné than in the southern rural area (18%, 95% CI: 15–22 *vs.* 6%, 95%CI: 3–10) or in the northern rural area (18%, 95% CI: 15–22 *vs.* 5%, 95%CI: 4–8). Schistosomiasis was more prevalent among students (18%, 95% CI: 14–22) and those with no income-generating occupation (11%, 95% CI: 8–15).

## Distribution of STH and *Schistosoma* infection intensity

As presented in Table 4, the intensity of *T. trichiura* (81%) and hookworm (95%) infections was mostly light. *A. lumbricoides* infection was either light (44%) or moderate (44%) in intensity, whereas 35% of *S. haematobium* cases were classified as heavy infections with a greater proportion of heavy infections in the 1–4 (86%) and 50 and over (50%) age groups.

## Prevalence of protozoan infections

A total of 364 participants were infected with protozoan parasites (Fig 2). As presented in Table 5, the prevalence of *Plasmodium spp*. was 13% (142/1080, 95% CI: 11–15). Among the 142 participants infected with *Plasmodium* parasites, 28 (20%) presented with fever at the time of inclusion in the study, and 34 (24%) reported a history of fever over the three

**Table 3. Prevalence and distribution of filariasis and *Schistosoma haematobium* infection among the study population.**

| | Any filariasis | | | *Loa loa* | | *Mansonella perstans* | | *Schistosoma haematobium* | | |
|---|---|---|---|---|---|---|---|---|---|---|
| | N | n, (%) | 95%CI | n, (%) | 95%CI | n, (%) | 95%CI | N | n, (%) | 95%CI |
| **Overall** | 1076 | 94, (8.7) | 7.1 – 10.6 | 76, (7.1) | 5.6 – 8.8 | 30, (2.8) | 1.9 – 4.0 | 1017 | 115, (11.3) | 9.4 – 13.4 |
| **Age range** | | | | | | | | | | |
| 1 – 4 | 108 | 1, (0.9) | 0.0 – 5.0 | 1, (0.9) | 0.0 – 5.0 | 0, (0.0) | – | 98 | 7, (7.1) | 2.9 – 14.2 |
| 5 – 19 | 339 | 9, (2.6) | 1.2 – 5.10 | 8, (2.4) | 1.0 – 4.6 | 1, (0.3) | 0.0 – 1.6 | 326 | 66, (20.2) | 16.0 – 25.0 |
| 20 – 49 | 315 | 33, (10.5) | 7.3 – 14.4 | 25, (7.9) | 5.2 – 11.5 | 14, (4.4) | 2.4 – 7.3 | 298 | 32, (10.7) | 7.5 – 14.8 |
| ≥ 50 | 314 | 51, (16.2) | 12.3 – 20.8 | 42, (13.4) | 9.8 – 17.6 | 15, (4.8) | 2.7 – 7.8 | 295 | 10, (3.4) | 1.6 – 6.1 |
| **Gender** | | | | | | | | | | |
| Female | 575 | 38, (6.6) | 4.7 – 9.0 | 29, (5.0) | 3.4 – 7.2 | 9, (1.6) | 0.7 – 3.0 | 542 | 55, (10.1) | 7.7 – 13.0 |
| WRA* | 207 | 9, (4.3) | 2.0 – 8.1 | 5, (2.4) | 0.8 – 5.5 | 4, (1.9) | 0.5 – 4.9 | 200 | 29, (14.5) | 9.9 – 20.1 |
| Male | 501 | 56, (11.2) | 8.5 – 14.3 | 47, (9.4) | 7.0 – 12.3 | 21, (4.2) | 2.6 – 6.3 | 475 | 60, (12.6) | 9.8 – 16.0 |
| **Location** | | | | | | | | | | |
| Lambaréné | 479 | 23, (4.8) | 3.1 – 7.1 | 19, (4.0) | 2.4 – 6.1 | 4, (0.8) | 0.2 – 6.1 | 469 | 85, (18.1) | 14.7 – 21.9 |
| Southern rural area | 220 | 40, (18.2) | 13.3 – 23.9 | 31, (14.1) | 9.8 – 19.4 | 16, (7.3) | 4.2 – 11.5 | 208 | 13, (6.2) | 3.4 – 10.4 |
| Northern rural area | 377 | 31, (8.2) | 5.6 – 11.5 | 26, (6.9) | 4.5 – 9.9 | 10, (2.6) | 1.3 – 4.9 | 340 | 17, (5.0) | 2.9 – 7.9 |
| **Occupation** | | | | | | | | | | |
| No occupation | 350 | 22, (6.3) | 4.0 – 9.4 | 22, (6.3) | 4.0 – 9.4 | 12, (3.4) | 1.8-5.9 | 329 | 37, (11.2) | 8.0 – 15.2 |
| Farmer | 252 | 43, (17.1) | 12.6 – 22.3 | 36, (14.3) | 10.2 – 19.2 | 16, (6.3) | 3.7 – 10.1 | 236 | 9, (3.8) | 1.8 – 7.1 |
| Student | 383 | 8, (2.1) | 0.9 – 4.1 | 8, (2.1) | 0.9 – 4.1 | 1, (0.3) | 0.0 – 1.4 | 363 | 64, (17.6) | 13.9 – 21.9 |
| Other | 91 | 21, (23.1) | 14.9 – 33.1 | 10, (11.0) | 5.4 – 19.3 | 1, (1.1) | 0.0 – 6.0 | 89 | 5, (5.6) | 1.8 – 12.6 |

*Women of reproductive age.

past days from the day of inclusion. A total of nine intestinal protozoan species were detected, as depicted in Fig 2. Table 5 presents their overall distribution in the study population, with an overall prevalence of 28% (266/944, 95% CI: 25–31). As presented in S1 Table, the most-common species were *Blastocystis hominis* (11%, 101/944, 95%CI: 9–13) and *Entamoeba coli* (9%, 80/944, 95%CI: 7–10). *Iodamoeba bütschlii* was detected in 4% (95%CI: 2.9–5.5) of our participants, whereas *Endolimax nana* (31, 944), *Giardia duodenalis* (27/944) and *Entamoeba histolityca/dispar* (27/944) were each detected in 3% (95% CI: 2–4) of our participants. Only seven cases of *Cyclospora cayetanensis*, three cases of *Cystoisospora belli*, and three cases of *Chilomastix mesnili* were detected.

## Distribution of protozoan infections in the study population

Table 5 presents the distributions of *Plasmodium* spp. and intestinal protozoans in the study population. *Plasmodium* spp. were more prevalent among participants aged 5–19 years (21%, 95%IC: 17–26), males (15%, 95% CI: 11–18), students (18%, 95% CI: 14–22), and participants living in southern rural area (26%, 95% CI: 20–32). Intestinal protozoa were less prevalent in participants aged 1–4 years (17%, 95% CII: 10–26), with no difference between males (30%, 95% CI: 26–35) and females (26%, 95% CI: 22–30). A significantly higher prevalence was observed among those living in southern rural area (36%, 95% CI: 29–43) than among those living in northern rural area (23%, 95% CI: 19–28). As presented in S1 Table, *Entamoeba coli* was more prevalent in people aged 50 years and over (12%, 95% CI: 8–16) and in those living in southern rural area (15%, 95% CI: 10–21) than in Lambaréné (4%, 95% CI: 2–7). Similarly, *Iodamoeba butschlii* was more prevalent in people living in southern rural area (9%, 95% CI: 6–14), whereas *Blastocystis hominis* was more prevalent among participants living in Lambaréné (15%, 95% CI: 12–19).

**Table 4. Prevalence and distribution of soil-transmitted helminth and *Schistosoma haematobium* infection intensities among the study population.**

| | Ascaris lumbricoides: n, (%) | | | | Trichuris trichiura: n, (%) | | | | Hookworm: n, (%) | | | | S. haematobium: n, (%) | | |
|---|---|---|---|---|---|---|---|---|---|---|---|---|---|---|---|
| | N | Light | Moderate | Heavy | N | Light | Moderate | Heavy | N | Light | Moderate | Heavy | N | Light | Heavy |
| **Overall** | 45 | 20, (44.4) | 20, (44.4) | 5, (11.2) | 113 | 92, (81.4) | 17, (15.0) | 4, (3.6) | 82 | 87, (94.6) | 4, (4.3) | 1, (1.1) | 115 | 75, (65.2) | 40, (34.8) |
| **Age range** | | | | | | | | | | | | | | | |
| 1–4 | 2 | 1, (50.0) | 0, - | 1, (50.0) | 6 | 5, (83.3) | 0, - | 1, (16.7) | 0 | 1, (100.0) | 0, - | 0, - | 7 | 1, (14.3) | 6, (85.7) |
| 5 – 19 | 26 | 10, (38.5) | 14, (53.8) | 2, (7.7) | 40 | 26, (65.0) | 13, (32.5) | 1, (2.5) | 13 | 13, (100.0) | 0, - | 0, - | 66 | 44, (66.7) | 22, (33.3) |
| 20 – 49 | 5 | 2, (40.0) | 3, (60.0) | 0, - | 34 | 31, (91.2) | 2, (5.9) | 1, (2.9) | 23 | 2, (100.0) | 0, - | 0, - | 32 | 26, (81.2) | 6, (18.8) |
| ≥ 50 | 11 | 6, (54.5) | 3, (27.3) | 2, (18.2) | 33 | 30, (90.9) | 2, (6.1) | 1, (3.0) | 46 | 41, (89.1) | 4, (8.7) | 1, (2.2) | 10 | 5, (50.0) | 5, (50.0) |
| **Gender** | | | | | | | | | | | | | | | |
| Female | 25 | 13, (52.0) | 10, (40.0) | 2, (8.0) | 50 | 31, (62.0) | 16, (32.0) | 3, (6.0) | 28 | 25, (89.3) | 3, (10.7) | 0, - | 55 | 36, (65.5) | 19, (34.5) |
| WRA* | 5 | 2, (40.0) | 3, (60.0) | 0, - | 12 | 8, (66.7) | 3, (25.0) | 1, (8.3) | 3 | 3, (100.0) | 0, - | 0, - | 29 | 23, (79.3) | 6, (20.7) |
| Male | 20 | 7, (35.0) | 10, (50.0) | 3, (15.0) | 63 | 61, (96.8) | 1, (1.6) | 1, (1.6) | 54 | 52, (96.3) | 1, (1.9) | 1, (1.9) | 60 | 39, (65.0) | 21, (35.0) |
| **Location** | | | | | | | | | | | | | | | |
| Lambaréné | 16 | 4, (20.0) | 8, (60.0) | 4, (20.0) | 17 | 33, (82.4) | 3, (17.6) | 0, - | 12 | 12, (100) | 0, - | 0, - | 85 | 50, (58.8) | 35, (41.2) |
| Southern rural area | 20 | 10, (50.0) | 10, (50.0) | 0, - | 26 | 26, (100.0) | 0, - | 0, - | 34 | 31, (91.2) | 3, (8.8) | 0, - | 13 | 11, (84.6) | 2, (15.4) |
| Northern rural area | 9 | 6, (45.0) | 2, (35.0) | 1, (20.0) | 70 | 33, (74.3) | 14, 20.0) | 4, (5.7) | 46 | 44, (95.7) | 1, (2.2) | 1, (2.2) | 17 | 14, (82.4) | 3, (17.6) |
| **Occupation** | | | | | | | | | | | | | | | |
| No occupation | 11 | 6, (50.0) | 4, (40.0) | 1, (10.0) | 18 | 16, (88.9) | 2, (11.1) | 0, - | 13 | 12, (92.3) | 1, (7.7) | 0, - | 30 | 21, (70.0) | 9, (30.0) |
| Farmers | 10 | 5, (44.4) | 3, (33.3) | 2, (22.2) | 38 | 35, (92.1) | 2, (5.3) | 1, (2.6) | 57 | 53, (93.0) | 3, (5.3) | 1, (1.8) | 8 | 6, (75.0) | 2, (25.0) |
| Students | 23 | 8, (34.8) | 13, (56.5) | 2, (8.7) | 45 | 29, (64.4) | 13, (28.9) | 3, (6.7) | 13 | 13, (100) | 0, - | 0, - | 63 | 40, (63.5) | 23, (36.5) |
| Others | 1 | 1, (100.0) | 0, - | 0, - | 12 | 12, (100.0) | 0, - | 0, - | 9 | 9, (100.0) | 0, - | 0, - | 14 | 8, (57.1) | 6, (42.9) |

- Zero percent, *Women of Reproductive Age.

**Table 5. Prevalence and distribution of the main protozoa in the study population.**

| | *Plasmodium* spp. | | | Any intestinal protozoa | | |
|---|---|---|---|---|---|---|
| | N | n, (%) | 95%CI | N | n, (%) | 95%CI |
| Overall | 1080 | 142, (13.1) | 11.2 – 15.3 | 944 | 266, (28.2) | 25.3 – 31.2 |
| **Age range** | | | | | | |
| 1 – 4 | 108 | 017, (15.7) | 09.4 – 24.0 | 088 | 015, (17.0) | 09.9 – 26.5 |
| 5 – 19 | 343 | 072, (21.0) | 16.8 – 25.7 | 306 | 098, (32.0) | 26.8 – 37.6 |
| 20 – 49 | 315 | 027, (08.6) | 05.7 – 12.2 | 266 | 066, (24.8) | 19.7 – 30.4 |
| ≥ 50 | 314 | 026, (08.3) | 05.5 – 11.9 | 284 | 087, (30.6) | 25.3 – 36.3 |
| **Gender** | | | | | | |
| Female | 577 | 068, (11.8) | 09.3 – 14.7 | 515 | 135, (26.2) | 22.5 – 30.2 |
| WRA* | 208 | 016, (07.7) | 04.5 – 12.2 | 183 | 043, (23.5) | 17.6 – 30.3 |
| Male | 503 | 074, (14.7) | 11.7 – 18.1 | 429 | 131, (30.5) | 26.2 – 35.1 |
| **Location** | | | | | | |
| Lambaréné | 480 | 041, (08.5) | 06.2 – 11.4 | 405 | 117, (28.9) | 24.5 – 33.6 |
| Southern rural area | 223 | 059, (26.5) | 20.8 – 32.8 | 194 | 069, (35.6) | 28.8 – 42.7 |
| Northern rural area | 377 | 042, (11.1) | 08.1 – 14.8 | 345 | 080, (23.2) | 18.8 – 28.0 |
| **Occupation** | | | | | | |
| No occupation | 290 | 039, (13.4) | 09.7 – 17.9 | 240 | 058, (24.2) | 18.9 – 30.1 |
| Farmer | 249 | 024, (09.6) | 06.3 – 14.0 | 230 | 072, (31.3) | 25.4 – 35.5 |
| Student | 379 | 069, (18.2) | 14.4 – 22.5 | 337 | 102, (30.3) | 25.4 – 35.5 |
| Others | 162 | 010, (06.2) | 03.0 – 11.1 | 137 | 034, (24.8) | 17.8 – 32.9 |

*Women of Reproductive Age, *Histolytica/Dispar*

## Geographical distribution of helminths and protozoa

The geographical distribution of helminths reveals relatively uniform distribution of cases of *T. trichiura* and hookworm infections across the study areas, whereas cases of *A. lumbricoides* and *S. stercoralis* infections are less common and may be completely absent in certain areas distant from the N1 national road, such as Nvey, Bindo and Saint Martin for *A. lumbricoides*, and Nvey for *S. stercoralis* (Fig 3). *S. haematobium* cases were also concentrated in Lambaréné and over some kilometres along the N1 national road from the northern edge of the town. Filarial infections with *L. loa* were present throughout the study areas, although only to a very limited extent, whereas *M. perstans* was confined to Lambaréné and villages on the N1 national road south of the town (Fig 4). In contrast to helminths, the distribution of protozoans did not show any clear geographical pattern, either for intestinal protozoa or *Plasmodium* spp. (Fig 5).

## Helminths and protozoan coinfections

Our analysis of coinfection between the ten protozoan species and seven helminth species revealed 341 cases of mono-infections (58%, 95% IC: 54–62) and 244 cases of co-infections (42%, 95% IC: 38–46). The distribution of co-infection is detailed in S2 Table. Among the coinfected individuals, 152 (62%, 95% IC: 56–68) were infected with two parasite species, 68 (28%, 95% IC: 22–34) with three species, and 16 (7%, 95% IC: 4–10) with four species. Only six participants (2%, 95% IC: 1–5) were infected with five species, and two participants (1%, 95% IC: 0–3) were infected with six species. Among the 244 cases of co-infection, the most frequently found species among helminths were *T. trichiura*, hookworm and *S. haematobium* with 79 (32%, 95% CI: 26–39), 76 (31%, 95% CI: 25–37), and 57 (23%, 95% CI: 18–29) occurrences, respectively, whereas among protozoa, *Plasmodium* spp., *Blastocystis hominis* and *Entamoeba coli* were

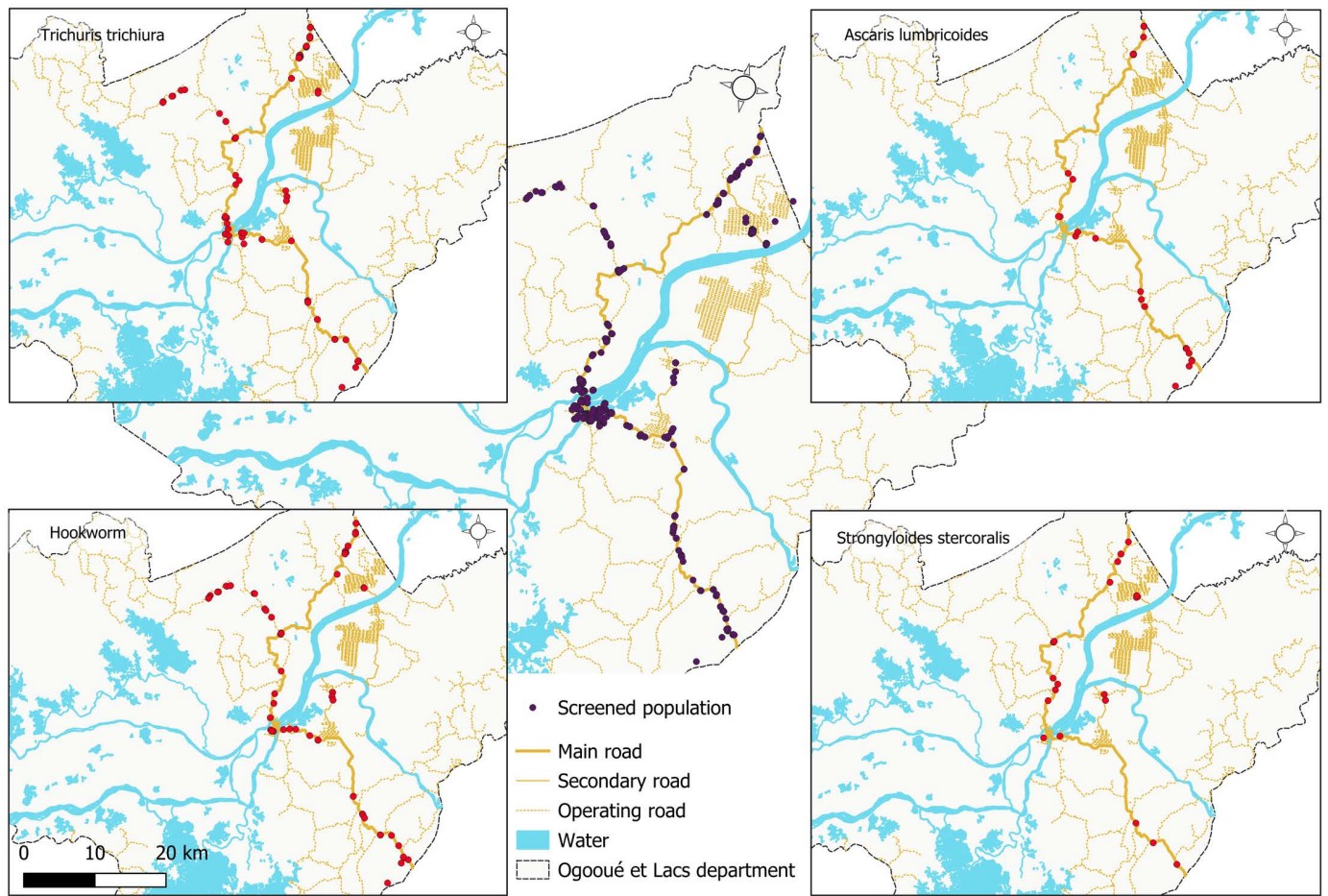

**Fig 3. Geographical distribution of soil-transmitted helminth infection cases.** The base layer for the country and the shape of the national borders were obtained from the 2018-2022 GADM database; an open licence (https://gadm.org/download_country.html). QGIS version 3.34.11 was used to create the map.

the most common species involved with 79 (32%, 95% CI: 26–39), 53 (22%, 95% CI: 18–27), and 50 (20%, 95% CI: 16–26) occurrences, respectively.

## Discussion

The objective of our study was to estimate the prevalence of helminth and protozoan infections in CERMEL study areas of Moyen-Ogooué. As shown in Fig 6, the survey identified several species of helminths and protozoans within the study areas. Overall, one out of two participants was infected with either helminths and/or protozoa. Specifically, 33% of the study population was infected with any helminth species, including STHs (21%), schistosomiasis (11%), and filariasis (9%), whereas 24% was infected with protozoa, including intestinal protozoa (28%) and *Plasmodium* spp. (13%).

With a 33% prevalence, Moyen-Ogooué Province can be considered a moderate-risk area for helminth infections, with STH being the most common. The most prevalent STHs were *T. trichiura* (12%) and hookworm (9%), followed by *A. lumbricoides* (5%) and *S. stercoralis* (3%). Our findings align with previous studies conducted in various regions of Gabon, which also identified *T. trichiura* as the most prevalent STH species [24,31]. However, these results contrast with studies from Ribado-Meñe et al. in Equatorial Guinea [32] or Kaboré et al. in Congo [33], where *A. lumbricoides* and hookworm

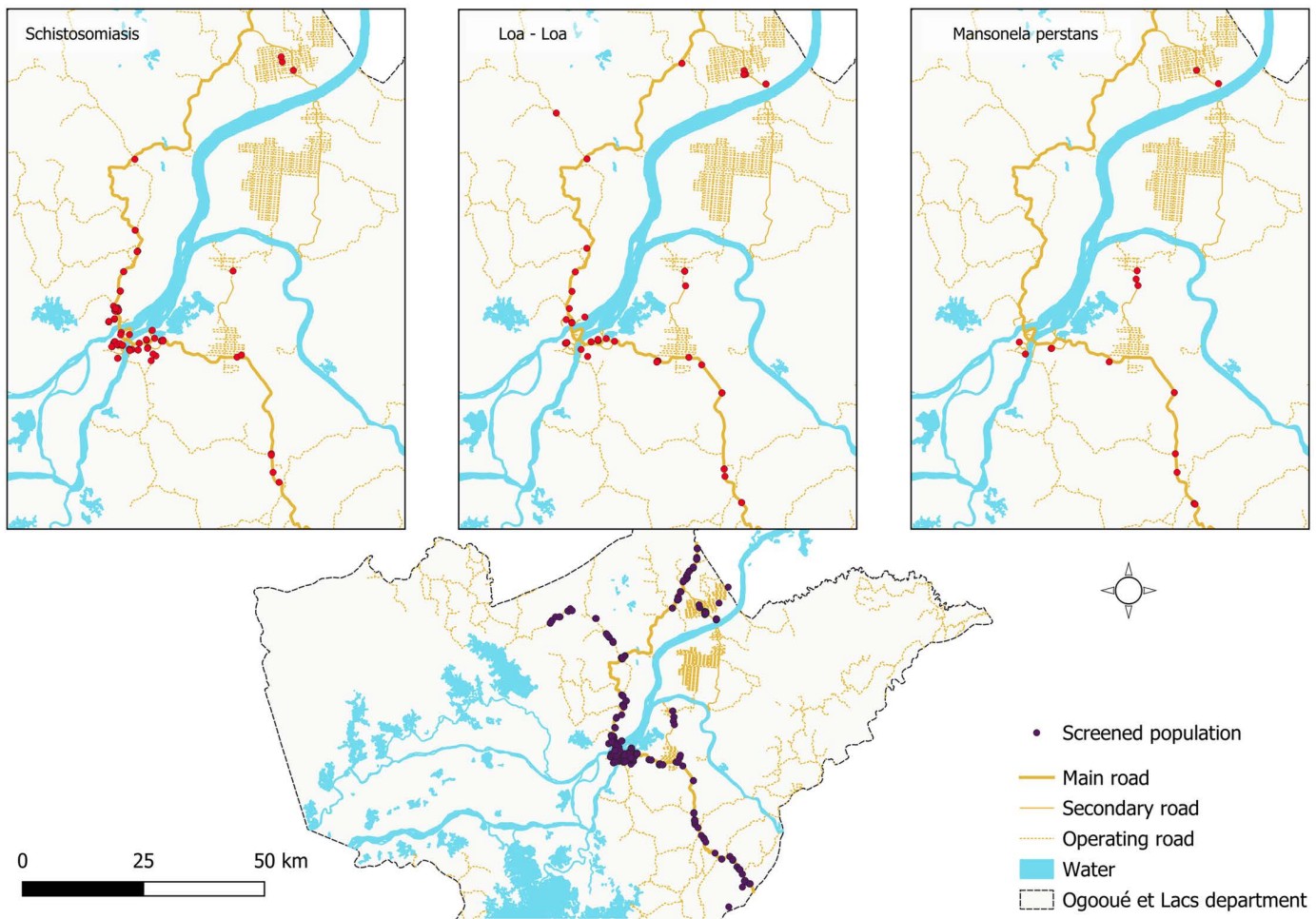

**Fig 4. Geographical distribution of *Loa loa* and *Mansonella perstans* infection cases.** The base layer for the country and the shape of the national borders were obtained from the 2018-2022 GADM database; an open licence (https://gadm.org/download_country.html). QGIS version 3.34.11 was used to create the map.

were the predominant STH species, respectively. This discrepancy could be explained by the widespread use of anthelminthics such as ABZ or mebendazole (MBZ) in Gabon, either as self-medication or through MDA programs. Indeed, 51% of our study population reported using anthelminthics at least once in previous years. While benzimidazole-based drugs are highly effective against *A. lumbricoides* and hookworms, with an egg reduction rate of over 95%, even at a single dose, they are less effective against *T. trichiura* and *S. stercoralis* [34,35], possibly explaining the higher prevalence of these species. In fact, Ribado-Meñe et al. attributed the high prevalence of STHs in Equatorial Guinea in part to the absence or few implementations of large-scale campaigns of ABZ or MBZ in the country [32].

The low prevalence of schistosomiasis in the present study (11%, 95%CI: 9–13) differs from the moderate prevalence (26%, 95%CI: 22–29) we reported in the same region in 2016 [24]. The difference could be due to the broader age range of the participants included in the present study (1–97), as well as the wide study area, compared with the previous study, which included participants under 18 years of age and was conducted only in Lambaréné. This hypothesis is supported by the fact that in the present study, the 5–12 and 13–19 age groups living in Lambaréné presented a higher schistosomiasis prevalence (27%, 95%CI: 21–34), similar to that reported in 2016, suggesting that children and

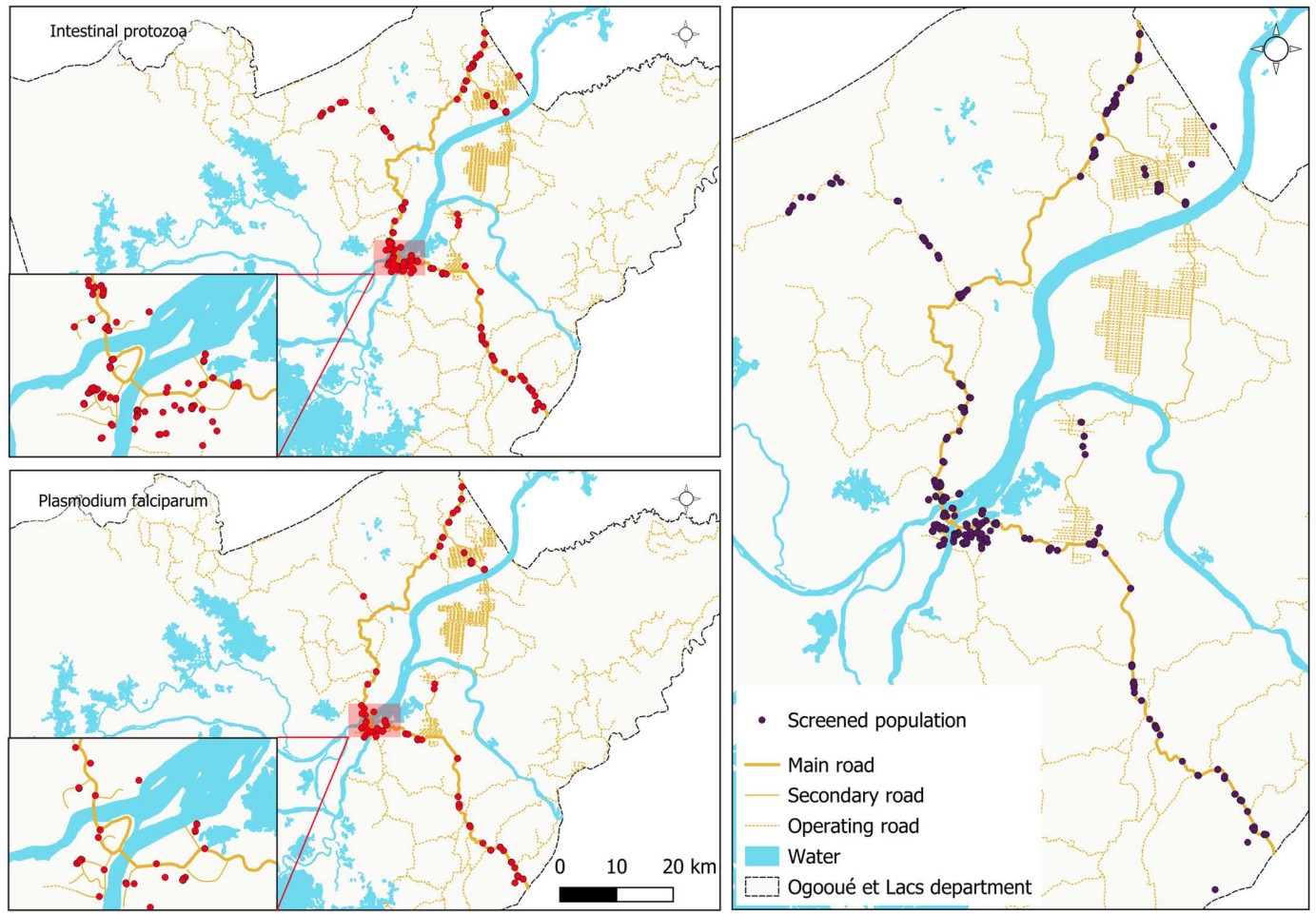

**Fig 5. Geographical distribution of intestinal protozoa and _Plasmodium_ spp. infection cases.** The base layer for the country and the shape of the national borders were obtained from the 2018-2022 GADM database; an open licence (https://gadm.org/download_country.html). QGIS version 3.34.11 was used to create the map.

teenagers, particularly those from Lambaréné, carry the greatest burden of the disease, with the morbidity of the disease remaining consistent in that population over time. Another relevant finding was that 14% (95%CI: 10 – 20) of WRA were infected with _S. haematobium_. Considering that 75% of women with urinary schistosomiasis are positive for female genital schistosomiasis [36], we can assume that at least one out of ten WRA in our community is affected by FGS, which should be a concern.

When assessing the intensity of helminth infections, including STH and schistosomiasis, we generally detected mild to moderate intensities of all the STH species. However, one-third of schistosomiasis cases were classified as heavy infections. Our findings for STH infections are consistent with other studies from endemic areas [14,16]. As suggested by Ribado-Meñe et al., who reported a similar profile in terms of the infection intensity of STH infection in Equatorial Guinea, despite the high prevalence of these infections observed in the country [32], high-intensity infections are typically treated due to the symptoms they cause, prompting individuals to self-medicate with anthelminthic or seek medical advice. As described in the literature [37,38], the severe cases we observed were mainly among elderly individuals, mainly farmers, and school-aged children. The relatively low proportion of severe schistosomiasis cases in our study may be explained by

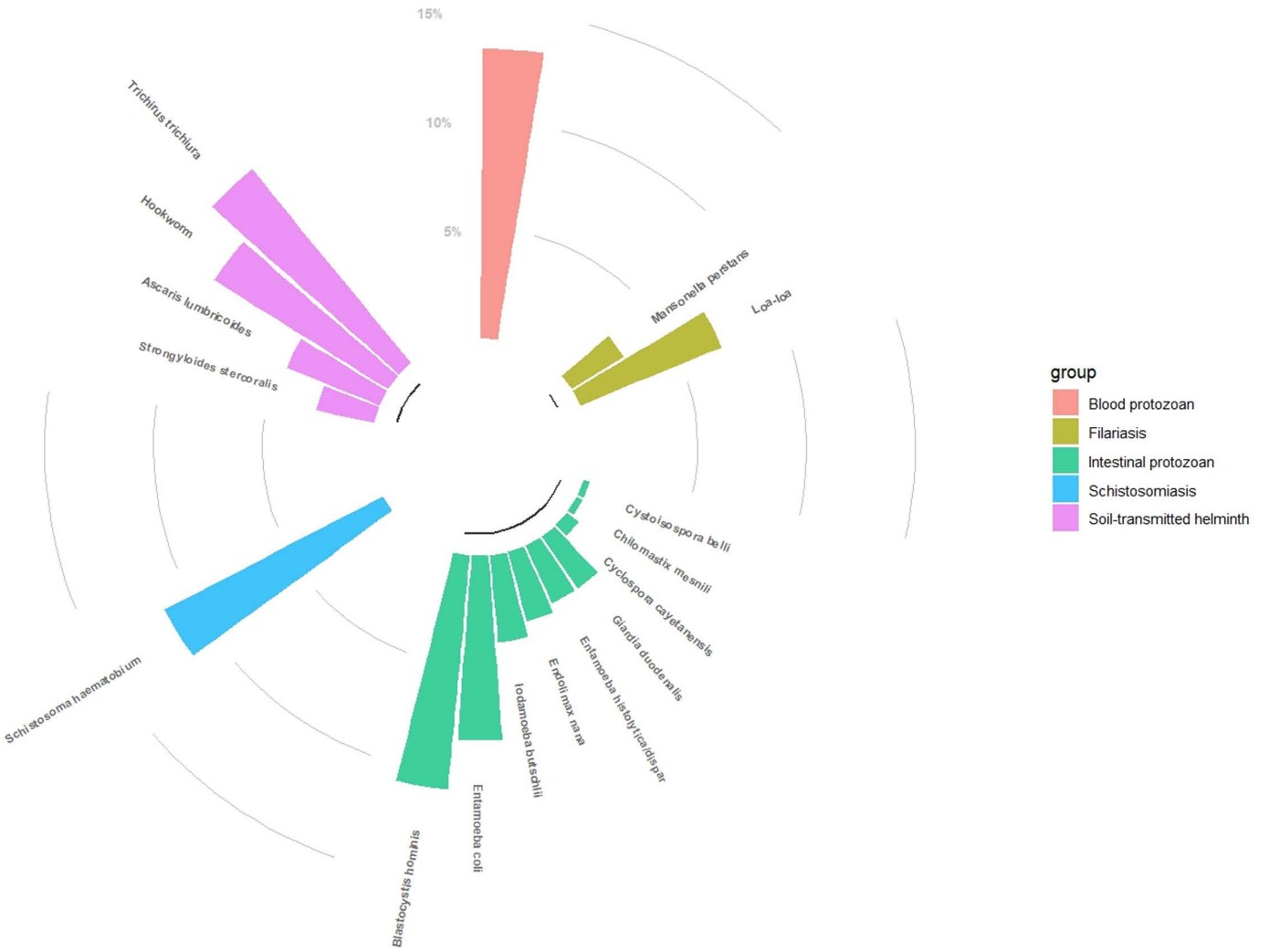

**Fig 6. Circular barplot of helminths and protozoa found during the survey.**

the use of praziquantel in the community. A recent report from the same study area indicated that one-third of inhabitants had received praziquantel in the past [39]. Although MDA campaigns for ABZ, MBZ, and praziquantel are not common in our community, the use of these anthelminthics in the population as a result of self-medication likely contributes to the control of STH and schistosomiasis morbidity. This practice should be encouraged.

We investigated the distribution of helminth infection throughout the study area: Our findings reveal different geographical distribution patterns of helminth species. If an absence of effective treatment can explain the uniform distribution of cases of *T. trichiura*, the same pattern found for hookworm could highlight a similar risk of transmission for hookworm across the study areas. The focal distribution of ascariasis and probably strongyloidiasis sometimes totally absent in some rural areas can be supported by the fact that, in such an areas, "contaminated" foods are less transmitted through the population than it can be done in Lambaréné, the semi-urban area characterised by the presence of markets, or outlets selling prepared foods such as salads or sandwiches. Surprisingly, *S. haematobium* cases were concentrated in Lambaréné, which can be explained by the observed migration of populations, particularly children, from southern rural areas

formerly known as highly endemic for schistosomiasis [40], to Lambaréné. Although the distribution of the forest and its proximity to households could explain the distribution of filariasis, the absence of *M. Perstans* species north of Lambaréné is questionable, as well as the absence of distribution pattern observed for protozoans.

We found a 9% (95%CI: 7–10) prevalence of filariasis, which is consistent with a report of M'bondoukwé et al. for another town in Gabon [17] but lower than the prevalence reported by Hemilembolo et al. in the Republic of Congo [41]. The lower prevalence in our study may be attributed to our study population, as filariasis risk increases with age [42,43] due to prolonged exposure to vectors and occupations such as manual labour [42]. The long lifespan of the worm [44] and the absence of safe treatment options contribute to the increased prevalence and intensity of infection in adults. Our results revealed a marked increase in prevalence with age, reaching 16% in participants aged 50 years and older, which is consistent with findings from Congo [41]. Interestingly, while *M. perstans* infections began at age 19, *Loa loa* infections were detected as early as the age of four years, suggesting closer contact between the population and the vectors (*Chrysops silacea* or *C. dimidiata*) that transmit *Loa loa* than to the midges of the genus *Culicoides,* which is responsible for transmitting *M. perstans* filariasis.

Protozoal infections, particularly intestinal protozoa and *Plasmodium* spp., were the second focus of our study. We identified nine intestinal protozoan species (see Fig 2), with *Blastocystis hominis* and *Entamoeba coli* being the most prevalent. If the latter is known to be nonpathogenic, *B. hominis* is sometimes associated with digestive symptoms – although there is broad consensus that it is apathogenic, too. As its clinical relevance remains unclear, further investigations could be needed to understand the role of *B. hominis* in global morbidity in the community. As intestinal protozoan species of clinical relevance, we found, although at a low prevalence (3%), *E. histolytica/dispar* and *G. duodenalis* but also three cases of *Cystoisospora belli* infection known to particularly affect immunocompromised people. We did not highlight a particular pattern of the distribution of these intestinal protozoa in our study population, particularly with respect to age, gender, and occupation. The epidemiological profile we reported here is similar to findings from previous studies in Gabon [17], Côte d'Ivoire [45]. To the best of our knowledge, this is the first time that the distribution of protozoan species has been reported for the Moyen–Ogooué province, providing valuable data for managing parasitic infections in the region and complementing a previous study published in 2018 conducted in two other provinces of the country [17]. This could help provide a better description of the epidemiological situation of these protozoa in the country.

We found an overall prevalence of 13% (95%CI: 11–15) for *Plasmodium* spp., which was lower than the 19% reported in southwestern Gabon [17] and the 28% reported for an area close to the province of Moyen-Ogooué [46]. The difference may be attributed to the study duration and population. Our study spanned 13 months and likely included more low-prevalence seasons, whereas the study conducted by M'bondoukwé et al. had a shorter duration of nine months. Additionally, the inclusion of pregnant women in the Fougamou study, a neighbouring town, likely contributed to their higher prevalence. Nonetheless, our results highlight that children aged 5–12 years and teenagers bear the greatest burden of *Plasmodium* infection, making them a priority population for infection control efforts. Indeed, because they are among the vulnerable populations for malaria, much attention is paid to preventing the disease in children under 5 years of age, to the detriment of those aged 5–12 years, for example. The difference in prevalence that we observed between the two age groups underlies this fact. In addition, more than half (56%) of the participants positive for *Plasmodium* parasites presented with no fever or history of fever. This could indicate a high level of asymptomatic infection in our community and deserves further investigation.

Coinfections with multiple parasite species were common in our study, with some participants infected with as many as five or six species. This finding mirrors the results of M'bondoukwé et al. in 2014, who reported similar rates of coinfection in two other provinces of Gabon [17]. This finding indicates that several parasitic infections, particularly *Plasmodium* spp., *Schistosoma* spp. and *T. trichiura,* which are the most common coinfections, could share the same risk factors such as environmental factors. The high rate of coinfection raises questions about comorbidities, particularly since interactions between schistosomiasis and *Plasmodium* infection may be influenced by the presence of STHs, such as hookworms and

*T. trichiura* [47]. Multiple parasitic infections may increase susceptibility to malaria, as has been observed in Gabonese populations [17]. Eight participants were coinfected with five or six parasite species. Although they were mainly male (8) and from rural area (9), either farmer (4) or student (4), their relative low number does not allow for the investigation of the underlying reasons. Further investigations will be welcome.

Although each participant was only seen once during the survey, data collection spanned 13 months. We recognize that this could influence the distribution of certain parasites, particularly *Plasmodium* spp., whose prevalence may vary seasonally. However, as *Plasmodium* transmission occurs year-round [48], our data provide valuable insight into the geographical distribution of infections and their co-occurrence with other parasitic species.

## Conclusion

This study revealed a moderate prevalence of helminths and protozoa in our community, with age, gender, and location playing a significant role in the infection distribution depending on the parasite. Coinfection between helminths and protozoa is common, particularly involving *T. trichiura* and *Plasmodium* spp. Further investigation into the interactions between these species could provide valuable insights for controlling helminth transmission in our region.

## Supporting information

**S1 Table. Distribution of protozoan parasitic species found during the survey, with the exception of *Cyclospora cayetanensis*, *Cystoisospora belli* and *Chilomastix mesnili* which were found in very low numbers.**
(DOCX)

**S2 Table. List of combinations of helminth and protozoan coinfections.**
(DOCX)

**S1 Dataset. Excel sheets containing all data used for statistical analysis (right sheet) and the codebook (left sheet).**
(XLSX)

## Acknowledgments

We would like to take this opportunity to thank all our participants in this study and the parents of our minor participants, without whom this achievement would not have been possible. We would also like to thank all the CERMEL laboratory teams (parasitology and clinical laboratories), as well as the field team, particularly the drivers, the nurses (Moussavou Saguiliba Hilda Scholastique, Tchibinda Bibalou Rebecca Debora Zoé, and Bibina Mihindou Jessica), and the field workers (Kengué Mbamba Daniella, Bissiélou John Steve, and Massandé Jean-Aimé) for their dedication to their work. AAA, MPG and JCDA are members of the EDCTP-NoE CANTAM.

## Author contributions

**Conceptualization:** Jean Claude Dejon Agobé, Ayôla Akim Adegnika.

**Data curation:** Jean Claude Dejon Agobé, Christian Chassem-Lapue, Ynous Djida.

**Formal analysis:** Jean Claude Dejon Agobé, Christian Chassem-Lapue, Mahmoudou Saidou, Ynous Djida.

**Funding acquisition:** Peter Gottfried Kremsner, Ayôla Akim Adegnika.

**Investigation:** Jean Claude Dejon Agobé, Christian Chassem-Lapue.

**Methodology:** Jean Claude Dejon Agobé, Ayôla Akim Adegnika.

**Project administration:** Jean Claude Dejon Agobé.

**Supervision:** Ayôla Akim Adegnika.

**Validation:** Ayôla Akim Adegnika.

**Visualization:** Jean Claude Dejon Agobé, Ayôla Akim Adegnika.

**Writing – original draft:** Jean Claude Dejon Agobé, Christian Chassem-Lapue.

**Writing – review & editing:** Jean Claude Dejon Agobé, Paul Alvyn Nguema-Moure, Moustapha Nzamba Maloum, Roméo-Aimé Laclong Lontchi, Jean Ronald Edoa, Yabo Josiane Honkpéhèdji, Jeannot Fréjus Zinsou, Bayodé Roméo Adégbitè, Martin Peter Grobusch, Peter Gottfried Kremsner, Ayôla Akim Adegnika.

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
