## [Decision Letter · Decision Letter 0]

PNTD-D-25-00265Community surveys of the prevalence, distribution, and coinfection of helminth and protozoan infections in semiurban and rural areas of Gabon, Central AfricaPLOS Neglected Tropical Diseases  Dear Dr. Dejon Agobé, Thank you for submitting your manuscript to PLOS Neglected Tropical Diseases. After careful consideration, we feel that it has merit but does not fully meet PLOS Neglected Tropical Diseases's publication criteria as it currently stands. Therefore, we invite you to submit a revised version of the manuscript that addresses the points raised during the review process. Please submit your revised manuscript within 30 days Jun 23 2025 11:59PM. If you will need more time than this to complete your revisions, please reply to this message or contact the journal office at plosntds@plos.org. Please include the following items when submitting your revised manuscript: * A rebuttal letter that responds to each point raised by the editor and reviewer(s). You should upload this letter as a separate file labeled 'Response to Reviewers '. This file does not need to include responses to any formatting updates and technical items listed in the 'Journal Requirements' section below. * A marked-up copy of your manuscript that highlights changes made to the original version. You should upload this as a separate file labeled 'Revised Manuscript with Track Changes '. * An unmarked version of your revised paper without tracked changes. You should upload this as a separate file labeled 'Manuscript '. If you would like to make changes to your financial disclosure, competing interests statement, or data availability statement, please make these updates within the submission form at the time of resubmission. Guidelines for resubmitting your figure files are available below the reviewer comments at the end of this letter. We look forward to receiving your revised manuscript. Kind regards, Maria AlmeriaGuest EditorPLOS Neglected Tropical Diseases Shaden Kamhawi

co-Editor-in-Chief

Paul Brindley

co-Editor-in-Chief

**Additional Editor Comments (if provided):** Please see reviewer 1's comments. In addition, please respond to the Scientific editor comments:

Scientific editor comments:The manuscript “Community surveys of the prevalence, distribution, and coinfection of helminth and protozoan infections in semiurban and rural areas of Gabon, Central Africa” by Dejon Agobé et al. is a cross-sectional study of 1,084 participants, aimed to describe the distribution of helminth and protozoan infections in the Moyen-Ogooué province of Gabon, where there is near 70,000 inhabitants. The results indicated a moderate prevalence of helminths and protozoa in that community, with age, gender, and location playing a significant role in the parasitic distribution. Although some parasite intensity of infection was mostly light, the intensity of Schistosoma haematobium was heavy in 35% of the cases, which is of concern, particularly when affected women of reproductive age that can have female genital schistosomiasis. A high number of different helminths and protozoa (3 participant were infected with six different species), parasites were observed in coinfections. It is the first study in that province in Gabon. Previous studies were performed in other provinces of Gabon (references 24 and 31, line 429, also reference 17). It is of interest that lower parasites seem to be related to the widespread use of anthelmintics in the country. The cross-sectional study was conducted with great care, including sample size calculation and sampling procedures.

Line 132-133. Did the authors analyze the presence of parasites in relation to the rainy seasons (2) or dry seasons (2)?

Line 137. Clarify that the semi urban area mentioned for Lambarene is area 1 of the study.

Line 147. The considered prevalence of 50% was higher than that observed. Would that modify the number of samples needed? Since the study collected much more than the samples needed, this is just a comment for personal interest.

Section on laboratory analysis, could give some more details on the specific methods used. As indicated all the analysis were based only on microscopy.

Any staining techniques performed to visualized acid-alcohol resistant protozoa?

Line 183- Thet text mentions “coproculture” for hookworms. Were any hookworms identified to the genera/species level?

Line 182. Spell out TBS.

Statistical consideration. The tests used for statistical analysis could be indicated. E.g. anova, t-test, Ch-square….

Line 236. Were the two cases of Enterobius vermicularis related to age?

Tables. Authors could add an asterisk in those parasites and factors that were statistically significant to easier understanding of differences.

Titles of lines 245, 260 and 272. Please change “Distribution” by “Prevalence”.

Lines 256-257 and lines 257-258. Please include p-values in the results statistically significant. Same for other sections.

Lines 278 and 280. “More prevalent” but statistically significant?

Line 283. Modify “trichuriasis” by “trichiura”

Line 285. The heaviest infections by S. haematobium seemed to occur in age 1-4 years and older than 50 years. Is this correct? Please indicate in text.

Line 309. Higher prevalence, but statistically significant?

Line 322. Where filarial infections found in areas where their vectors are more frequently found?

The discussion is sometimes a repetition of the results and could be reduced.

Line 396. Change “aera” by “area”.

Line 449. Indicate the common risk factors for those parasites.

The title of Suplementary table S1 do not correspond to the data shown in that table. **Journal Requirements:**

1) Some material included in your submission may be copyrighted. According to PLOSu2019s copyright policy, authors who use figures or other material (e.g., graphics, clipart, maps) from another author or copyright holder must demonstrate or obtain permission to publish this material under the Creative Commons Attribution 4.0 International (CC BY 4.0) License used by PLOS journals. Please closely review the details of PLOSu2019s copyright requirements here: PLOS Licenses and Copyright. If you need to request permissions from a copyright holder, you may use PLOS's Copyright Content Permission form.

Potential Copyright Issues:

- Figures: 1, 3, 4, and 5. Please (a) provide a direct link to the base layer of the map (i.e., the country or region border shape) and ensure this is also included in the figure legend; and (b) provide a link to the terms of use / license information for the base layer image or shapefile. We cannot publish proprietary or copyrighted maps (e.g. Google Maps, Mapquest) and the terms of use for your map base layer must be compatible with our CC BY 4.0 license.

2) In the online submission form, you indicated that "Data are available from the Centre de Recherce Medicale de Lambarene(email:admin@cermel.org) for researchers who meet the criteria for access to confidential data.". All PLOS journals now require all data underlying the findings described in their manuscript to be freely available to other researchers, either

1. In a public repository

2. Within the manuscript itself

3. Uploaded as supplementary information.

3) Please amend your detailed Financial Disclosure statement. This is published with the article. It must therefore be completed in full sentences and contain the exact wording you wish to be published. Please ensure that the funders and grant numbers match between the Financial Disclosure field and the Funding Information tab in your submission form. Note that the funders must be provided in the same order in both places as well.

**Reviewers' comments:** Reviewer's Responses to Questions

**Key Review Criteria Required for Acceptance?**

**Methods**

-Are the objectives of the study clearly articulated with a clear testable hypothesis stated?

-Is the study design appropriate to address the stated objectives?

-Is the population clearly described and appropriate for the hypothesis being tested?

-Is the sample size sufficient to ensure adequate power to address the hypothesis being tested?

-Were correct statistical analysis used to support conclusions?

-Are there concerns about ethical or regulatory requirements being met?

Reviewer #1: The objectives are clearly stated which is to describe the distribution of helminth and protozoan infections in the Moyen-Ogooue province of Gabon with a clear hypothesis. The study design was appropriate to address the stated objective.

The population was clearly defined as “all individuals aged one year and older living in the study areas for at least one year”. What was the exclusive criteria for the study?

The population size, was appropriate. A minimum size of 816 participants was obtained after sample size calculation. Correct Descriptive statistical analysis was used and confidence intervals to estimate precision of prevalence rates to support conclusions. Ethical requirements were met. No new analysis/experiments are required.

**Results**

-Does the analysis presented match the analysis plan?

-Are the results clearly and completely presented?

-Are the figures (Tables, Images) of sufficient quality for clarity?

Reviewer #1: The analysis presented match the analysis plan. The results are well presented. Most of the Figures are clearly presented except for Figure 6.

**Conclusions**

-Are the conclusions supported by the data presented?

-Are the limitations of analysis clearly described?

-Do the authors discuss how these data can be helpful to advance our understanding of the topic under study?

-Is public health relevance addressed?

Reviewer #1: The data presented supports the conclusion which shows moderate prevalence of helminths. Interpretation of prevalence of Protozoans should be in the context of regional epidemiological data stating the percentage. The authors have clearly discussed the importance of these data and its public health relevance.

**Editorial and Data Presentation Modifications?**

Reviewer #1: Authors to correct the following:

Line 86- Remove ‘for example’ in the sentence

Line 93- spelling error – platyhelminths instead of plathelminths

Line 102- spelling error – mastigophora instead of mastigora

Line 119- correct the cause of filariasis. It is caused by Wuchereria bancrofti, Brugia malayi and Brugia timori not Onchocerca volvolus

Line 144- State the exclusive criteria in the study

In Table 1- Female to sex ratio should be 1:15 not 1.15

Line 228-229 – The prevalence of 36% (357/980, 95% CI: 18 – 23) NOT ‘IC’ should be stated in Fig 2 under the total number for helminths as this is not seen anywhere in the tables. This should also be done for Protozoans.

Line 231 – Insert Table 3- after (94/1084,95%CI: 7-10) Table 3.

Line 247 – Prevalence rate was rounded up to 8% instead of 8.3%. I suggest Prevalence rates be left as it is.

Line 265 – Prevalence rate is 9.4% not 8% as written

Line 266 – Prevalence rate of M. perstans is 4.2% for males vs 1.6% for females

Line 278 – Prevalence of 18% not 16%

Line 279 – Prevalence of 5% and not 16%

Line 291 – 293 There is no information in Table 5 that gives the data of 28 (20%) infected participants presenting with fever at the time of inclusion and 34 (24%) reporting a history of fever over the 3 past days from the day of inclusion. Is there a supporting document for this?

Line 294 – 300 talks about the different parasites as shown in Figure 2. Insert Fig 2 after the sentence in in line 294 for clarification.

Line 305 – 310 contains information in Table 5. Insert Table 5 to indicate this.

Line 330 – 341 has no Table or Figure to support the information given about Helminths and Protozoan coinfections. A table should be included for this.

Fig.6 is not clear. There is a need for a sharp and clearer figure

Line 499 – Insert the names of the authors in the reference

I recommend minor revision of the manuscript.

**Summary and General Comments**

Reviewer #1: The study has provided epidemiological data on helminth and protozoan infection in the community. Authors need to do minor corrections.

PLOS authors have the option to publish the peer review history of their article (what does this mean? ). If published, this will include your full peer review and any attached files.

**Do you want your identity to be public for this peer review?** For information about this choice, including consent withdrawal, please see our Privacy Policy .

Reviewer #1: **Yes: ** Lydia Etuk Udofia

---

## [Editor Report · Decision Letter 1]

Dear Dr Dejon Agobé,

We are pleased to inform you that your manuscript 'Community surveys of the prevalence, distribution, and coinfection of helminth and protozoan infections in semiurban and rural areas of Gabon, Central Africa' has been provisionally accepted for publication in PLOS Neglected Tropical Diseases.

Best regards,

Maria Almeria

Guest Editor

Jong-Yil Chai

Section Editor

Shaden Kamhawi

co-Editor-in-Chief

Paul Brindley

co-Editor-in-Chief

The authors made all the necessary changes, and the manuscript can now be considered acceptable for publication.

---

## [Editor Report · Acceptance letter]

Dear Dr Dejon Agobé,

We are delighted to inform you that your manuscript, "Community surveys of the prevalence, distribution, and coinfection of helminth and protozoan infections in semiurban and rural areas of Gabon, Central Africa," has been formally accepted for publication in PLOS Neglected Tropical Diseases.

Best regards,

Shaden Kamhawi

co-Editor-in-Chief

Paul Brindley

co-Editor-in-Chief
